# Reducing Implicit Bias in Latent Domain Learning

## Abstract

A fundamental shortcoming of deep neural networks is their specialization to a single task and domain. While recent techniques in multi-domain learning enable the learning of more domain-agnostic features, their success relies firmly on the presence of domain labels, typically requiring manual annotation and careful curation of datasets. Here we focus on *latent domain learning*, a highly realistic, yet less explored scenario: learning from data from different domains, without access to domain annotations. This is a particularly challenging problem, since standard models exhibit an implicit bias toward learning only the large domains in data, while disregarding smaller ones. To address this issue, we propose *dynamic residual adapters* that adaptively account for latent domains, and *weighted domain transfer* — a novel augmentation strategy designed specifically for this setting. Our techniques are evaluated on image classification tasks containing multiple unannotated domains, and we demonstrate they enhance performance, in particular, on the smallest of these.

## 1 Introduction

While the performance of deep learning has surpassed that of humans in a range of tasks (He et al., 2016; Silver et al., 2017), machine learning models perform best when the learning objective is narrowly defined. Practical realities however often require the learning of joint models over semantically different examples. In this case, best performances are usually obtained by fitting a collection of models, with each model solving an individual subproblem. This is somewhat disappointing seeing how humans and other biological systems are capable of flexibly adapting to a large number of scenarios (Kaiser et al., 2017).

Past solutions that address this problem tend to fall into some category of multi-domain learning (Nam & Han, 2016; Bulat et al., 2019; Schoenauer-Sebag et al., 2019). In this setting, models are learned over diverse datasets each associated with an underlying distribution. Multi-domain learning however relies firmly on the availability of domain annotations, for example to control domain-specific architectural elements (Rebuffi et al., 2017; 2018; Liu et al., 2019; Guo et al., 2019).

Reliance on domain annotations is not limited to the multi-domain scenario, their presence is also required in domain adaptation where models transfer between related tasks (Ganin et al., 2016; Tzeng et al., 2017; Hoffman et al., 2018; Xu et al., 2018; Peng et al., 2019a; Sun et al., 2019b), continual learning (Kirkpatrick et al., 2017; Lopez-Paz & Ranzato, 2017; Riemer et al., 2019), meta learning over multiple tasks (Finn et al., 2017; Li et al., 2018a), or the generalization to previously unseen domains (Li et al., 2018b; 2019b;a; Gulrajani & Lopez-Paz, 2020).

The above approaches have established the notion that the presence of domain labels improves generalization. In the real world however, these can often be difficult or expensive to obtain. Consider images that were scraped from the web. Most image datasets such as Pascal VOC (Li et al., 2018a) or ImageNet (Deng et al., 2009) already rely on expensive manual filtering and the removal of different looking images. Existing multi-domain approaches require that the scraped images are further annotated for the mixture of content types they will contain, such as real world images or studio photos (Saenko et al., 2010), clipart or sketches (Li et al., 2017). This can be an expensive process, moreover it is not clear which variations (indoor/outdoor, urban/rural, etc.) should be grouped.

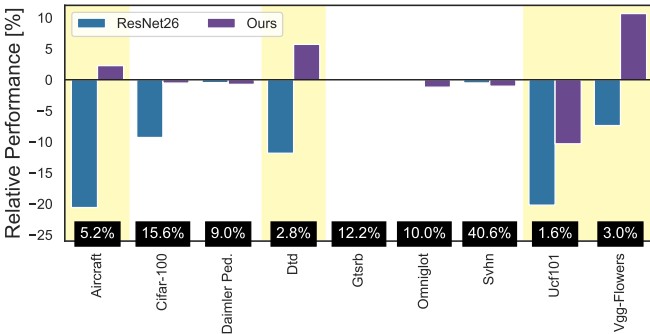

Figure 1: Changes in performance without access to domain labels, relative to the multi-domain baseline of learning an individual baseline on each dataset (sizes in %). For ResNet26 (•), performance losses on the smallest domains (yellow) are significant. Our proposed solutions (•) recover a large portion of the gap to full domain-supervison. Best viewed in color.

Here we consider the task of learning in absence of domain labels, or *latent domain learning*. This scenario encompasses any task where we have inadequate resources to assign domain labels to all data, but have reason to believe that such a partitioning of the data would in principle make sense. And as our experiments show, even *when domain labels already exist, there is no guarantee that these are optimal* for a given model. In this paper, we therefore argue and demonstrate that learning implicit domain assignments, end-to-end alongside the rest of the network, is the superior option.

In Figure 1 we display accuracy for latent domain learning by jointly learning a single model over datasets from the Visual Decathlon benchmark that contains images from distinct domains with mutually exclusive classes. Per-domain performance is measured relative to that of 9× models learned individually on each domain — a common baseline in the multi-domain setting (Rebuffi et al., 2018; Liu et al., 2019). This highlights the central challenge of latent domain learning: a significant loss of performance for the joint model (•) on small domains. While the performance drop is not significant on large domains, relative accuracy is reduced by 15-20% on Aircraft and Ucf101, for example. Latent domain learning therefore requires customized solutions, because standard models have *an implicit bias through which they disregard the smallest domains in data*.

The mechanisms we propose throughout this paper (shown in •) help overcome this: *dynamic residual adapters* (Section 3.3), which we couple with *weighted domain transfer* (Section 3.4) obtain robust performance on small domains, without trading performance on larger ones. Our proposed solutions can be incorporated and trained seamlessly with existing architectures, and are able to surpass the performance of domain-supervised approaches that have access to human-annotated domain labels (Section 4.2). Moreover, qualitative analysis demonstrates that they partition latent domains in a highly intuitive way (Figures 2 and 3).

## 2 RELATED WORK

Multi-domain learning relates most closely to our paper. The state-of-the-art introduces small convolutional corrections in residual networks to account for individual domains (Rebuffi et al., 2017; 2018). Stickland & Murray (2019) extend this approach to obtain efficient multi-task models for related language tasks. Other recent work makes use of task-specific attention mechanisms (Liu et al., 2019), attempts to scale task-specific losses (Kendall et al., 2018), or addresses tasks at the level of gradients (Chen et al., 2017). Crucially, these approaches all rely firmly on domain labels.

A lack of domain labels has previously attracted interest in domain adaptation, Hoffman et al. (2012) use hierarchical clustering to uncover latent domains, other work investigates the use of kernel-based clustering (Gong et al., 2013), via exemplar SVMs (Xu et al., 2014), or mutual information (Xiong et al., 2014). Different from these works, we propose tackling latent domains in an end-to-end fashion, which no longer requires a clustering ansatz. In another line of work Mancini et al. (2018) estimate batch statistics of domain adaptation layers with Gaussian mixture models using only few domain labels. Peng et al. (2019b) study the shift from some source domain to a target distribution

that contains multiple latent domains. In our setting, there is no significant shift between source and target distributions, instead the focus lies on learning parameter efficient models that generalize well across multiple latent domains.

Furthermore, our work is loosely related to learning universal representations (Bilen & Vedaldi, 2017), which Tamaazousti et al. (2019) use as a guiding principle in designing more transferable models. However, these works also assume the presence of domain labels. Multi-modal learning does not make this assumption: Deecke et al. (2018) normalize data in separate batches to account for differences in feature distributions, while Chang et al. (2018) propose an architecture that accounts for latent semantic factors to match images. As we show in our experiments, latent domain learning benefits from more customized solutions than these.

Our module gives rise to a differentiable dynamic network architecture, studied e.g. in the context of reinforcement learning (Zoph & Le, 2016; Pham et al., 2018), Bayesian optimization (Kandasamy et al., 2018), when adapting to new tasks (Mallya et al., 2018; Rosenfeld & Tsotsos, 2018), or in universal object detection (Wang et al., 2019). For such architectures, two components are commonly used: Gumbel-softmax sampling (Jang et al., 2016), e.g. leveraged in dynamic computer vision architectures (Veit & Belongie, 2018; Sun et al., 2019a), or mixtures of experts (Jacobs et al., 1991; Jordan & Jacobs, 1994), used to scale models to large problem spaces (Shazeer et al., 2017).

From the perspective of algorithmic fairness, a desirable property for models is to ensure consistent predictive equality across different identifiable subgroups in the data (Zemel et al., 2013; Hardt et al., 2016; Fish et al., 2016; Corbett-Davies et al., 2017). This relates to the central goal in latent domain learning: to limit the implicit model bias toward large domains, and improve robustness on small domains. Wang et al. (2020) explore this connection for visual recognition problems, different from our work they focus their analysis on a setting in which domain annotations are available.

## 3 METHOD

### 3.1 PROBLEM SETTING

In multi-domain learning (Nam & Han, 2016; Rebuffi et al., 2017; 2018; Bulat et al., 2019; Schoenauer-Sebag et al., 2019), it is assumed that data is sampled i.i.d. from some mixture of domain distributions $\mathbb{P}_d$ with domain labels $d = 1, \ldots, D$. Together, they constitute the underlying distribution as $\mathbb{P} = \sum_d \pi_d \mathbb{P}_d$, where each domain is associated with a relative share $\pi_d = N_d/N$, with $N$ the total number of samples, and $N_d$ those belonging to the $d$'th domain. In multi-domain learning, the domain label $d$ is always available.

While the two are closely related, in the broader multi-task scenario (He et al., 2017; Kokkinos, 2017; Vandenhende et al., 2020) the nature of underlying tasks $t = 1, \ldots, \mathcal{T}$ is inherently different, and learning on each task distribution $\mathbb{P}_t$ is associated with an individual loss function $L_t$ (for example, one task may be object classification, the other semantic segmentation). In multi-domain learning on the other hand, all the $L_t$ are associated with an equivalent problem type, but can vary substantially depending on domain membership.[1] This is also the case for latent domain learning, with the important distinction that learning occurs over non-annotated domains.

In latent domain learning the information that associates each sample with a domain $d$ is no longer available. As such, we cannot infer domain-specific labels $y_d$ from sample-domain pairs $(\boldsymbol{x}, d)$ and are instead forced to learn a single model $f_{\boldsymbol{\theta}}$ over the entire $\mathbb{P}$. We study two potential cases for latent domain learning. The first setting (Section 4.1) includes mutually exclusive classes (as in Visual Decathlon) and a disjoint label space $\mathcal{Y}_1 \cup \cdots \cup \mathcal{Y}_D$ that encompasses all domains. This is a challenging setting, as models do not have a priori information about each domain's label space $\mathcal{Y}_d$, and may therefore assign samples to false ones. For the second one (Section 4.2), we focus on a setting in which label spaces are shared (for example, elephants can appear as a photo or painting) and $\mathcal{Y}_d = \mathcal{Y}_{d'}$ for all domains.

A standard baseline in multi-domain learning is to finetune $D$ individual models, one for each domain (Rebuffi et al., 2018; Liu et al., 2019). Doing so requires learning a large number of parameters

---

[1]In addition, in multi-task learning samples are often associated with a set of task labels $y_1, \ldots, y_{\mathcal{T}}$. In multi-domain learning, each sample has a distinct label $y_d$ with mutually exclusive $\mathcal{Y}_d$ for each domain $d$.

and shares no parameters across domains, but can serve as an upper guide on performance. We show that in some cases, even when domain annotations were assigned carefully, a latent domain approach can surpass the performance of such strong domain-supervised baselines, see Section 4.2.

## 3.2 RESIDUAL ADAPTATION

Residual adaptation builds on the observation that features from large pretraining tasks are universally applicable (Bilen & Vedaldi, 2017), and require only small modifications to correct for domain-specific differences. Rebuffi et al. (2017) use this insight to extend the layer-wise transformation of the widely adopted residual architecture (He et al., 2016):

$$\boldsymbol{x} + f_{\boldsymbol{\theta}}(\boldsymbol{x}) + h_{\boldsymbol{\alpha},d}(\boldsymbol{x}), \tag{1}$$

where $f_{\boldsymbol{\theta}}$ denotes the main convolutions of the residual network (with parameters $\boldsymbol{\theta}$), and $h_{\boldsymbol{\alpha},d}$ are small domain-specific convolutional corrections, i.e. $\dim(\boldsymbol{\alpha}) \ll \dim(\boldsymbol{\theta})$. While several concepts can in principle be incorporated with latent domains (Perez et al., 2018; Park et al., 2019), residual adaptation stands out due to its methodological simplicity and is a well-established method in the multi-domain literature.

In this work access to $d$ is removed, resulting in two new challenges: we have no a priori information about the right number of corrections $\{h_{\boldsymbol{\alpha},d}\}$, and we cannot use $d$ to decide which one of these to apply. Throughout the next section, we present an alternative strategy of inspecting $\boldsymbol{x}$ and choosing relevant corrections $h_{\boldsymbol{\alpha}}$ on the fly.

## 3.3 DYNAMIC RESIDUAL ADAPTERS

While there is no access to domain labels $d$ in latent domain learning, we still assume $\mathbb{P}$ is constituted by several domain distributions $\mathbb{P}_d$. To account for these in a domain-unsupervised fashion, we propose the use of *dynamic residual adapters* (DRA): each incoming sample $\boldsymbol{x}$ is first processed by $K$ corrections $\{h_{\boldsymbol{\alpha}_k}\}_{k=1,\dots,K}$, which we parametrize with light-weight 1x1 convolutions. Next, a gating mechanism $g_k$ is responsible for weighing each correction $h_{\boldsymbol{\alpha}_k}$, under which $\boldsymbol{x}$ is then transformed. In the $l$'th layer of the network, the subsequent feature representation computes as

$$\text{DRA}(\boldsymbol{x}) \triangleq \boldsymbol{x} + f_{\boldsymbol{\theta}_l}(\boldsymbol{x}) + \sum_{k=1}^{K} g_{lk}(\boldsymbol{x}) h_{\boldsymbol{\alpha}_{lk}}(\boldsymbol{x}), \tag{2}$$

with $g_{lk}(\boldsymbol{x})$ the $k$'th component of the $l$'th gating function. For an illustration, see Figure 4. While we motivate DRA from learning on latent domains, there is no guarantee that each gate will correspond to a single domain and many additional factors (shape, pose, color, etc.) may enter the gate assignments as well.

We parametrize the gating units using a self-attention mechanism (Lin et al., 2017). The only learnable parameters are a small linear transformation $W : \mathcal{C} \to \mathbb{R}^K$ s.t. $g(\boldsymbol{x}) = \text{Softmax}\{W^{\intercal}\phi(\boldsymbol{x}) + \boldsymbol{\varepsilon}\}$. Here $\phi(\boldsymbol{x})$ is a projection onto the channel dimension $\mathcal{C}$ that averages out height and width (average pooling), and $\boldsymbol{\varepsilon} \sim \mathcal{N}(0, \Sigma_{\varepsilon})$ a channel-wise noise that encourages gate exploration (Shazeer et al., 2017).[2] The gating mechanism corresponds to a categorical distribution over $K$ categories, i.e. $0 \le g_k \le 1$ and $\sum_k g_k = 1$. How to choose $K$ is discussed in more detail in Section 4.1.

While many other parametrizations of the gating function $g_k$ are possible, self-attention enables assignments and thus allows for the weighted combination of different $h_{\boldsymbol{\alpha}_k}$. Though discrete assignments can also be enforced through Gumbel-Softmax sampling (Jang et al., 2016), we found this approach to be too restrictive and underperforming. We compare different gate parametrizations in Table 5 of the Appendix.

## 3.4 WEIGHTED DOMAIN TRANSFER

The central challenge in latent domain learning is the tendency of models to overaccount for large domains in $\mathbb{P}$. Besides accounting for this via dynamic residual adapters, we introduce an augmentation technique that encourages information exchange between latent domains.

---

[2]Exploration noise is fixed at $\Sigma_{ii} = 10^{-2} \, \forall i$, zero otherwise.

We are motivated by the following example: assume two classes (say, cats and dogs) each with two latent domains (sketches and photos). Ideally, we would want to encourage the model to learn a domain-agnostic representation of $x$, from which it may infer $y$, invariant of its latent domain $d$. We achieve this here by augmenting $x$ with the style information of a second sample $x'$, drawn at random from $\mathbb{P}$ (so potentially, but not necessarily crossing domains).

A central challenge in latent domain learning is that we need to limit the transfer of information between highly similar samples, as this may restrict the model's access to discriminative features. For example, within the same domain we can often associate particular visual information with individual classes, e.g. aircraft tend to appear in front of a blue sky, while dogs appear in natural scenes. Conceptually speaking, if we transfer between these two samples too aggressively, we may change the background scene of the dog to sky, whereby the model will learn it can no longer correlate sky and aircraft. To stop this from happening, our methodology limits the transfer between samples that are too similar, but transfers when the visual information in them appears different.

In *weighted domain transfer* (WDT) we randomly pair each sample $x_i$ with a second sample $x_j$ from the mini batch. After average pooling samples onto channel dimensions s.t. $(\mu_i, \mu_j) \in \mathcal{C}$, we compute their pairwise similarity in terms of the Bhattacharyya distance $\delta_i = (\mu_i - \mu_j)^\mathsf{T} \sigma^{-1}(\mu_i - \mu_j) + 0.5 \ln(\det \sigma / \sqrt{\det \sigma_i \sigma_j})$.[3] Here $\sigma = (\sigma_i + \sigma_j)/2$ denotes the average of the channel-wise variances associated with each sample. Note distances here are scalar values, i.e. $\delta \in \mathbb{R}_+$.

Before exchanging information, distances have to be constrained to a suitable range. To achieve this, we normalize them with their associated mean $\mu_\delta = \sum_i \delta_i / n$ (where $n$ denotes the batch size) and standard deviation $\sigma_\delta$, squash them with a sigmoid $S$, and finally scale them with an exchange strength $\eta$, i.e. $\delta_{\eta i} = \eta S[(\delta_i - \mu_\delta)/\sigma_\delta]$. This yields a single exchange strength $\delta_{\eta i} \in [0, \eta]$ for each sample in the mini batch, which WDT uses to transfer information from $x_j$ to $x_i$:

$$x_{i,\text{new}} = \text{WDT}(x_i, x_j) \triangleq \delta_{\eta i}\left[\frac{\sigma_j(x_i - \mu_i)}{\sigma_i} + \mu_j\right] + (1 - \delta_{\eta i})x_i. \tag{3}$$

In practice, this combines the statistics of $x_i$ and $x_j$ with dynamic strength $\delta_{\eta i}$ depending on their pairwise similarity. Crucially for latent domain learning we do not want to transfer between samples that are very similar, so if their pairwise distance is small, then $\delta_{\eta i} \approx 0 \Rightarrow x_{i,\text{new}} \approx x_i$. Figure 2 (right) shows that $\delta_{\eta i}$ can vary substantially, and does so depending on the domain associated with $x_i$. In line with style exchange approaches (Gatys et al., 2016; Huang & Belongie, 2017), we find that applying WDT in early layers of the network is the most effective (see Table 7). Further ablations in Tables 5-6 show that WDT is stable under various choices of $\eta$ and significantly outperforms a competitive augmentation strategy MixUp (Zhang et al., 2017), which does not consider the similarity between samples and less suited for latent domains.

## 4 EXPERIMENTS

We consider two experimental settings to evaluate our proposed approaches: Visual Decathlon for a traditional multi-domain setting but without access to domain labels (Section 4.1), and a dataset that contains equivalent classes for multiple latent domains (Section 4.2). While not the focus of our paper, DRA also increases performance for single datasets and significantly improves robustness to class imbalance, see Tables 3 and 4 in the Appendix. All experiments were implemented in PyTorch (Paszke et al., 2017), and code will be made available alongside the camera ready version.

### 4.1 LATENT VISUAL DECATHLON

The first experiment combines nine datasets from the Visual Decathlon challenge (Rebuffi et al., 2017) that contain a variety of different images, with mutually exclusive labels, i.e. $|\mathcal{Y}| = \sum_d |\mathcal{Y}_d|$.[4] Note the goal here is not to compare to the performance of existing multi-domain approaches that

---

[3] While closely related, this metric has some improved stability properties relative to the Mahalanobis distance, which can be tricky to optimize in practice.

[4] For Visual Decathlon, this gives rise to a 2128-dimensional label space. This implies that in latent domain learning models may erroneously classify samples from one domain (say, SVHN) to another (e.g. Omniglot). In the multi-domain setting domain labels $d$ and their association to individual label spaces $\mathcal{Y}_d$ prevent this.

Table 1: Performances of $9\times$ ResNet26 learned individually on each domain versus that of latent domain models: ResNet26, ResNet56 and our DRA+WDT with $(K, \eta) = (2, 0.2)$. For domain-supervision ($9\times$ ResNet26), label space sizes vary between 2 (Daimler Pedestrian) and 1623 (Omniglot). In latent domain learning, there is a single label space over all domains $|\mathcal{Y}| = 2128$. Overall best performance underlined, best latent domain models in bold. Yellow: smallest domains.

| | Latent $d$ | Airc. | C-100 | Daim. | Dtd | Gtsrb | Omn. | Svhn | Ucf101 | Vgg-F. | W-Acc. |
|---|---|---|---|---|---|---|---|---|---|---|---|
| $\pi_d$ | | *0.052* | *0.156* | *0.09* | *0.028* | *0.122* | *0.1* | *0.406* | *0.016* | *0.03* | |
| $9\times$ResNet26 | ✗ | 39.48 | 77.96 | 99.95 | 38.19 | 99.95 | 87.62 | 95.12 | 73.00 | 65.20 | 87.01 |
| ResNet26 | ✓ | 31.35 | 70.71 | 99.49 | 33.67 | 99.87 | **87.80** | 94.64 | 58.25 | 60.39 | 84.73 |
| ResNet56 | ✓ | 34.62 | 71.63 | **99.52** | 34.79 | **99.90** | 87.72 | **95.12** | 60.66 | 57.55 | 85.22 |
| DRA+WDT ($K = 2$) | ✓ | **40.83** | 77.55 | 99.22 | 40.37 | 99.80 | 86.58 | 94.14 | 65.47 | 72.16 | 86.58 |
| DRA+WDT ($K = 3$) | ✓ | 39.36 | 78.27 | 99.10 | 41.97 | 99.80 | 86.35 | 94.21 | 63.83 | 71.96 | 86.62 |
| DRA+WDT ($K = 4$) | ✓ | 36.81 | 77.97 | 98.91 | 40.48 | 99.77 | 87.17 | 94.21 | 65.47 | 70.69 | 86.46 |
| DRA+WDT ($K = 5$) | ✓ | 40.38 | **78.64** | 99.03 | **42.29** | 99.77 | 86.60 | 93.94 | 63.93 | **72.94** | **86.68** |

Visual Decathlon was designed for, but to show that deep networks struggle with learning small latent domains when no domain annotations are provided.

For latent domain learning, domain labels are never used in training, and only serve to a posteriori decompose the weighted accuracy (denoted as W-Acc) reported in tables into domain-conditional metrics.

**Optimization** Initial ResNet parameters were obtained from ImageNet (Deng et al., 2009). For dynamic residual adapters, only gates and corrections are learned, the ResNet26 backbone remains fixed at its initial parameters. As Rebuffi et al. (2018) showed, this implicitly regularizes the network, while also benefiting performance (c.f. our ablation in Table 5). The exact same optimization routine is applied across all experiments: we trained for 120 epochs using stochastic gradient descent (momentum parameter of 0.9), batch size of 128, weight decay of $10^{-4}$, and initial learning rate of 0.1 (reduced by 1/10 at epochs 80, 100). We use the official splits for Visual Decathlon, and report average accuracies over five random initializations.

All experiments use standard augmentation techniques: random cropping and flipping, as well as normalization. For WDT, we experimented with different exchange amounts $\eta$. A range of values improve over having no augmentation, best results were obtained for $\eta = 0.2$, see Table 6. When increasing the number of residual adapters this also increases performance, however $K = 2$ already represents a good trade-off between learnable parameters and increase in modeling power.

**Results** In Table 1 we show results for a domain-supervised baseline of $9\times$ResNet26, one individual model for each domain $d$. We then learn a single ResNet26, this time as a latent domain model over all nine domains. Next, we couple DRA+WDT with the very same ResNet26. For further comparison, we also include a significantly deeper ResNet56.

Learning a single ResNet26 over latent domains with no access to labels significantly harms performance, with accuracy dropping significantly on the smaller domains. This problem is not addressed by simply increasing the depth of the network: while weighted accuracy improves slightly, ResNet56 exhibits the same implicit bias against small domains, leaking performance there. DRA+WDT close a large portion of the gap to domain-supervision and on four domains (Aircraft, Cifar-100, Dtd, Vgg-Flowers) even surpasses the performance of the domain-supervised baseline.

**Qualitative analysis** To understand better how DRA processes samples from different latent domains, it is helpful to inspect which gates activate across the entirety of the network. For a given sample, we follow its activation path across the $L$-layered network as $\Psi_k(\boldsymbol{x}) \triangleq (g_{k1}(\boldsymbol{x}), \ldots, g_{kL}(\boldsymbol{x}))$. When two samples have similar paths, this means they share similar residual corrections $h_{\boldsymbol{\alpha}_k}$. For simplicity, we set $K = 2$ in our analysis, s.t. $\Psi_1(\boldsymbol{x}) = 1 - \Psi_2(\boldsymbol{x})$.

We collected 26-dimensional gate activation paths for samples from all domains in Visual Decathlon, and visualize their principal components in Figure 2 (left). This reveals a highly intuitive clustering of domains: Svhn (•) and Gtsrb (•) both show real-world, yet strictly defined content (numbers and street signs). CIFAR-100 (•) is known to exhibit a large amount of variation, and correspondingly maps to a relatively large region – compare this to Omniglot (•) which is a visually much simpler

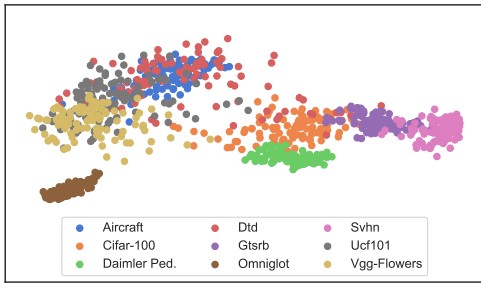 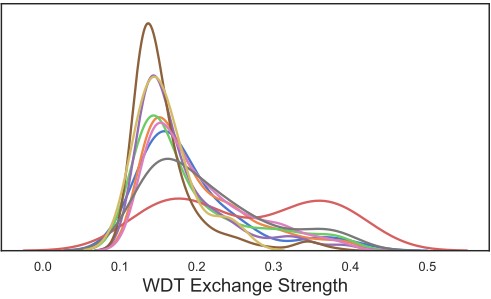

Figure 2: **Left**: PCA over gate activation paths. DRA partitions the data into separate latent domains in a semantically meaningful way: Omniglot (•) occupies a distinct region and also differs visually from the other datasets. Svhn (•) and Gtsrb (•) cluster together — both datasets contain real-world, but well-defined symbols (house numbers, street signs). **Right**: the effective distance $\delta_\eta$ that governs the amount of exchange in WDT for each latent domain. Omniglot shares little information with other domains, Dtd (•) shares more actively.

dataset, and occupies a more primitive manifold. Real-world datasets such as Ucf101 (•), Aircraft (•) and Dtd (•) share another common region.

These observations align with the exchange strength $\delta_\eta$ for WDT recorded in Figure 2 (right). There is little average exchange for Omniglot (•). Information exchange for Dtd (•) on the other hand is substantial, in correspondence with its gate activations that place it in the close proximity of other latent domains.

## 4.2 LATENT PACS

The second experiment examines performance on a dataset called PACS (Li et al., 2017), standing for its four constituting domains (*art painting, cartoon, photo, sketch*). Each domain contains samples of equivalent classes ("elephant", "giraffe", etc.). The domains are unbalanced (see $\pi_d$ in Table 2). We reserved 20% of samples for evaluation, leaving the remainder for training; we use random splits of the entire dataset, as the official splits are balanced for domains, whereas in the latent domain setting we assume no a priori knowledge of domain memberships. We make no changes to the optimization described in Section 4.1.

**Results** Table 2 shows that dynamic residual adapters improve considerably over the latent domain baseline of a joint ResNet26, with the largest gains on smaller domains (91.67% versus 85.27% on *art painting*). Notably, DRA surpasses the weighted accuracy of 4x domain-supervised ResNet26, while requiring a much smaller number of learnable parameters (3.5mil versus 24.8mil). Performance again increases with larger $K$ in DRA.

To the best of our knowledge, latent domain learning has not been targeted through customized deep learning architectures. A related baseline is MLFN (Chang et al., 2018), which builds on ResNeXt (Xie et al., 2017) to define a latent-factor architecture that accounts for multi-modality in data. Crucially, where we share small convolutional corrections at every layer, MLFN instead enables and disables entire network blocks, allowing us to outperform MLFN.

While domain-supervised residual adapters (Rebuffi et al., 2018) have been shown to work extremely well in the multi-domain scenario, their performance here is sub-par. This is likely because they exhibit no cross-domain sharing of parameters, which DRA does natively. As k-means (using $D = 4$ centers) with subsequent finetuning shows, obtaining fixed assignments across the depth of the network is suboptimal. In DRA we learn gates in an end-to-end fashion and flexibly share or separate features at every layer, thereby obtaining domain assignments that synergize with the overall model.

**Memory requirements** Adding DRA to the residual network results in additional memory requirements, since every layer requires $\mathcal{O}(K|\mathcal{C}| + K|\mathcal{C}|^2)$ learnable parameters to parametrize gates and corrections, respectively. As we outline in Table 2 this is an extremely modest increase, in par-

Table 2: Results on the PACS dataset. Shown are performances for ResNet26, MLFN, a domain-supervised ensemble of 4x ResNet26, and our DRA. Third column lists the number of parameters that have to be learned in each approach. Best overall performance underlined, best latent domain performance bold.

| | Latent $d$ | Param. [$\approx$] | Art Painting | Cartoon | Photo | Sketch | W-Acc. |
|---|---|---|---|---|---|---|---|
| $\pi_d$ | | | *0.205* | *0.235* | *0.167* | *0.393* | |
| RA (Rebuffi et al., 2018) | ✗ | 2.6 mil | 85.14 | 92.05 | 94.50 | 94.30 | 91.93 |
| 4x ResNet26 | ✗ | 24.8 mil | 88.41 | 95.53 | 94.34 | 95.71 | 93.94 |
| 4x ResNet56 | ✗ | 55.9 mil | 87.53 | 94.34 | 96.44 | 96.08 | 93.98 |
| k-means+RA | ✓ | 5.2 mil | 82.92 | 91.52 | 90.65 | 93.76 | 90.49 |
| ResNet26 | ✓ | 6.2 mil | 85.27 | 94.55 | 93.85 | 94.98 | 92.70 |
| ResNet56 | ✓ | 14.0 mil | 86.96 | 94.34 | 95.15 | 95.34 | 93.35 |
| MLFN (Chang et al., 2018) | ✓ | 7.6 mil | 78.38 | 91.29 | 88.19 | 92.95 | 88.78 |
| DRA+WDT ($K=2$) | ✓ | 1.4 mil | 90.46 | 94.77 | 97.41 | 94.73 | 94.31 |
| DRA+WDT ($K=3$) | ✓ | 2.1 mil | 90.10 | 94.88 | 97.09 | 95.16 | 94.38 |
| DRA+WDT ($K=4$) | ✓ | 2.8 mil | **91.67** | 95.42 | 96.60 | 95.10 | 94.72 |
| DRA+WDT ($K=5$) | ✓ | 3.5 mil | 90.10 | **95.86** | **97.41** | **95.40** | **94.76** |

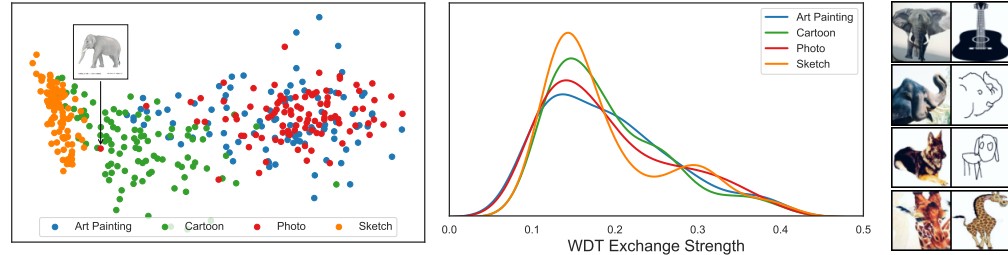

Figure 3: **Left**: PCA of samples represented by their $L$-dimensional activation paths. DRA shares parameters between visually similar domains *art painting* and *photo* (•,•), while isolating *sketch* (•). The arrow highlights one sample that has been labeled a *photo* in PACS. DRA categorizes it as a *cartoon* instead, a more adequate assignment for this particular elephant. **Middle**: WDT exchange for different domains, sketch (•) is particularly inactive. **Right**: samples from different domains with similar gate activations $\Psi_k$.

ticular as only a small subset of the parameters are learned (gates and residual corrections), while the convolutions of the residual backbone remain fixed.

**Qualitative analysis** As Figure 3 (left) shows, DRA exhibits an intuitive clustering of PACS domains: visually similar domains *art painting* and *photo* (•,•) cluster together. The manifold describing *sketches* (•) is arguably more primitive than those of the other domains, and indeed only maps to a small region. *Cartoon* (•) lies somewhere between sketches and real world imagery. This matches well with our intuition: a cartoon is, more or less, just a colored sketch. We highlight one particular elephant that is placed amongst the *cartoon* domain, but has been assigned a ground-truth domain label of *photo* in the PACS dataset. The ground-truth label was assigned in error, but different from approaches that use domain-supervision, dynamic residual adapters are not irritated by this. Exchange strengths in Figure 3 (middle) again confirms that WDT is more active for latent domains such as *art painting* and *photo* (•,•) that share parameters with each other.

As we demonstrate through samples from PACS in Figure 3 (right), similar activation paths $\Psi_k$ are indicative of additional visual similarities, not just domain membership: pose, color or edges of the shown samples are all closely related, compare in particular the poses of the elephants (second row).

Table 3: Accuracies for ResNet26, RA (Rebuffi et al., 2018), and our DRA.

| | ResNet26 | RA (Rebuffi et al., 2018) | DRA |
|---|---|---|---|
| CIFAR-10 | 95.20 | 95.80 | **96.32** |
| CIFAR-100 | 77.85 | 81.01 | **82.18** |

Table 4: Performance of ResNet26, ResNet56 and DRA on an unbalanced variant of CIFAR-10 where 75% of examples from classes 1-5 have been removed.

| Sample ratio | 0.25 | | | | | 1.0 | | | | | W-Acc. | Unif.-Acc. |
|---|---|---|---|---|---|---|---|---|---|---|---|---|
| $y$ | 1 | 2 | 3 | 4 | 5 | 6 | 7 | 8 | 9 | 10 | | |
| ResNet26 | 91.59 | 95.16 | 82.22 | 70.34 | 88.88 | 93.28 | 98.53 | 97.28 | 97.23 | 96.89 | 94.44 | 91.14 |
| ResNet56 | 90.03 | 93.30 | 84.97 | 74.25 | 87.55 | 93.96 | 98.33 | 97.83 | 97.81 | **97.45** | 94.86 | 91.65 |
| DRA | **93.43** | **96.39** | **91.09** | **79.96** | **93.59** | **95.59** | **99.15** | **98.45** | **98.65** | 97.10 | **96.45** | **94.44** |

## 4.3 SINGLE DATASETS

Dynamic residual adapters require no domain annotations, and can therefore be used for learning on single datasets. Table 3 contains test accuracies on CIFAR-10 and CIFAR-100 (Krizhevsky & Hinton, 2009) for DRA ($K = 2$). We compare this to standard finetuning of the residual backbone as well as residual adapters Rebuffi et al. (2018). DRA can be inserted seamlessly into the network, and we make no changes to the optimization outlined in Section 4.1.

On both datasets, dynamic residual adapters outperform traditional finetuning and residual adapters. We use no WDT here, which is specifically designed for the latent domain setting, but show that DRA can be used as a general purpose module to increase performance on benchmark datasets. Note the larger performance gap on CIFAR-100. In all likelihood DRA has a special advantage there, as CIFAR-100 is known to have small modes, which can be identified as small latent domains.

## 4.4 IMBALANCED DATA

The benefits of DRA are not restricted to the case of learning over multiple potentially unbalanced domains, but also extend to learning problems in which some classes are significantly underrepresented, i.e. the marginals $p(y) = \int_{\mathcal{X}} p(y|\boldsymbol{x})p(\boldsymbol{x})$ are heavily skewed. We construct the following experiment for CIFAR-10: samples $\boldsymbol{x}$ that belong to one of the first five classes (airplane, automobile, bird, cat, deer) are excluded from the dataset if $u < 0.25$ for $u \sim \mathcal{U}[0, 1]$, so that around 75% of them are removed. We learn models on this dataset using the standard optimization routine outlined in Section 4.1. Results are shown in Table 4.

Measured in uniform accuracy (which weighs performance equivalently for all classes), ResNet26 drops from 95.20% (c.f. Table 3) to 91.14%, because the model – similar to what was observed for latent domains – suppresses the underrepresented classes 1-5. Opposed to this, *DRA performs as well as it does on standard CIFAR-10 classification*. In other words, in the same way DRA raises the robustness for small latent domains, it also enhances models to have better performance on object classes for which less examples exist in the dataset.

We also display performance for ResNet56 (which has around 40% more parameters than DRA). Its poor performance highlights that DRA addresses a fundamental issue that off-the-shelf deep learning models exhibit when dealing with imbalanced distributions, and which cannot be resolved by simply learning a deeper network.

## 5 CONCLUSION

Recent work in multi-domain learning has been chiefly focused on a setting where domain annotations are assumed to be routinely available. As this requires careful curation of datasets, in real world scenarios this assumption is often of limited merit. Dynamic residual adapters inject adaptivity into networks, preventing them from overfitting to the largest domains in distributions, a failure mode of traditional models that is exposed in latent domain learning.

Not only does our approach close a large amount of the performance gap to domain-supervised solutions, but in some scenarios – even when domains have been assigned very carefully by human annotators – exceeds their performance. In the future, we hope to extend the mechanisms described here to the regime of extremely small domains, and examine additional research questions that arise there, e.g. whether they can help defend against adversarial attacks.

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

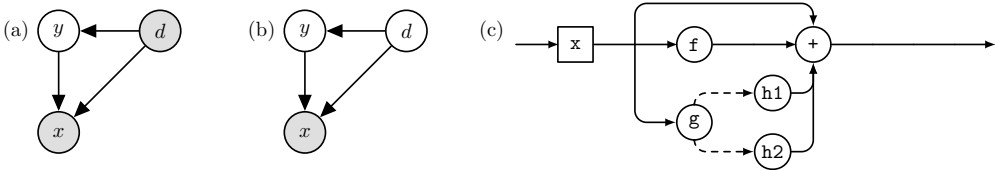

Figure 4: Graphical models for (a) multi-domain learning, (b) latent domain learning. (c) A ResNet block, equipped with a dynamic residual adapter ($K = 2$). Incoming samples x pass down three streams: an identity function, a transformation via a large convolution f, as well as an evaluation by expert gates g, which dynamically assigns (dashed arrows) small corrections h1 and h2.

---

**Algorithm 1** Single iteration of DRA and WDT.

---

**Input:** Parameters $(\eta, K)$, sample pairs $\{(\boldsymbol{x}_i, y_i)\}$. Fixed model parameters $\boldsymbol{\theta}$, learnable $\boldsymbol{\alpha}$, loss $\ell \colon \mathcal{Y} \times \mathcal{Y} \to \mathbb{R}_{\geq 0}$, learning rate $\gamma \in \mathbb{R}_{>0}$.

**for** each $x_i$ **do**

    $x_i \leftarrow \text{WDT}(\boldsymbol{x}_i, \boldsymbol{x}_j; \eta)$   `# draw `$x_j$` uniformly at random`

    $y_i' = f_{\boldsymbol{\alpha}|\boldsymbol{\theta}}(\boldsymbol{x}_i; K)$

    $\ell_i = \ell(y_i, y_i')$

**end for**

    $\boldsymbol{\alpha} \leftarrow \boldsymbol{\alpha} - \gamma \nabla_{\boldsymbol{\alpha}} \left[ \frac{1}{N} \sum_i \ell_i \right]$

**Return:** Latent domain model $f_{\boldsymbol{\alpha}|\boldsymbol{\theta}} \colon \mathcal{X} \to \mathcal{Y}$.

---

## A   APPENDIX

ILLUSTRATIONS

Figure 4 visualizes the difference between (a) multi-domain learning and (b) latent domain learning, in which domain labels $d$ remain hidden from the model. In (c) we visualize the routing within our proposed dynamic residual adapters.

ALGORITHM

In Algorithm 1 we outline pseudo-code for DRA and WDT. Note DRA can be implemented with standard models in a seamless way, and only modifies its internal structure. The resulting $f_{\boldsymbol{\alpha}\boldsymbol{\theta}}$ can be therefore trained in end-to-end fashion with any loss function through standard empirical risk minimization.

ABLATION

Our ablations show that latent domain learning benefits from both the addition of multiple dynamic residual adapters (i.e. $K > 1$) as well as WDT augmentation between feature maps of samples. Each row in Table 5 corresponds to changing a single parameter or design element for DRA or WDT, and is compared to the baseline configuration used in Table 1.

Removing the dynamic element of DRA by fixing a single residual adapter ($K = 1$) registers a significant performance loss relative to $K = 2$. In line with what Rebuffi et al. (2017) report, when not fixing parameters $\boldsymbol{\theta}$ of the ResNet convolutions, this leads to problems with overfitting. Replacing mixtures of experts with Gumbel-softmax sampling negatively impacted performance, as smooth interpolations between residual corrections are beneficial for sharing information between latent domains. Performances for soft and straight-through Gumbel-softmax sampling were on par, and we report straight-through sampling here.

Table 5: An ablation study on Visual Decathlon. First row contains DRA+WDT with default parameters $(K, \eta) = (2, 0.2)$. We remove each of our proposed solutions, and also replace it with an alternative parametrization: for DRA, this means that we remove the dynamic element and set a single ($K = 1$) residual adapter in each layer. Next we set $K = 2$, but replace the gating mechanism with Gumbel-Softmax sampling. Besides the complete removal of WDT in the fourth row, we also measure performance with MixUp as an alternative augmentation strategy. We also show accuracy when the residual network that we couple DRA+WDT with is not fixed at its pretrained parameters, a regularization strategy initially proposed in Rebuffi et al. (2017) that we also employ here.

| # | Fixed $\boldsymbol{\theta}$ | WDT | MixUp | DRA | DRA-Gumbel | W-Acc. | $\Delta$ |
|---|---|---|---|---|---|---|---|
| ours | ✓ | ✓ | | ✓ | | 86.58 | — |
| no gates (K=1) | ✓ | ✓ | | ✗ | | 85.65 | (-1.07%) |
| Gumbel gates | ✓ | ✓ | | ✗ | ✓ | 85.76 | (-0.95%) |
| no WDT | ✓ | ✗ | | ✓ | | 86.38 | (-0.23%) |
| MixUp | ✓ | ✗ | ✓ | ✓ | | 83.47 | (-3.59%) |
| unfix ResNet26 | ✗ | ✓ | | ✓ | | 85.35 | (-1.42%) |
| no DRA, no WDT | ✗ | ✗ | | ✗ | | 84.73 | (-2.14%) |

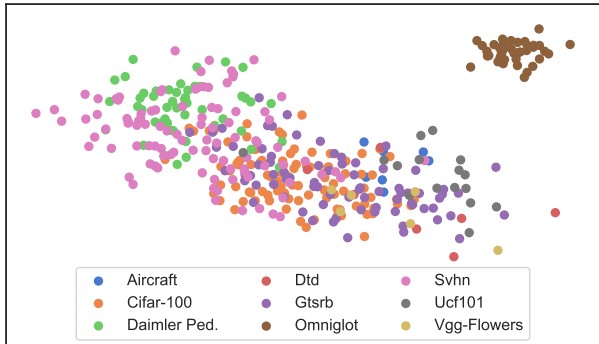

Figure 5: PCA over ResNet26. Only Omniglot is separated from other latent domains.

Having no augmentation from WDT ($\eta = 0$) does not impact weighted accuracy much (-0.23%), but drastically reduces performance on small domains: accuracy drops from 40.83% to 37.50% for Aircraft (-8.16%), as well as for Dtd (-2.23%), Ucf101 (-3.91%), and Vgg-Flowers (-2.99%).

MixUp (Zhang et al., 2017), an alternative augmentation that interpolates between samples, is not equally well suited for latent domain learning.

We include an additional qualitative analysis in Figure 5. This shows the embedding space learned by a ResNet26 without domain annotations. The model clearly separates Omniglot due to its low-level differences from other domains, but otherwise does not account for latent domain structure.

TUNING WDT

Results for additional choices of $\eta$ are shown in Table 6: most values for $\eta$ increase performance, when augmenting too little this however prevents a sufficient amount of exchange, whereas too strong an augmentation impairs the relationship between latent domains and their associated labels.

Table 6: Weighted accuracies for WDT under different exchange strengths $\eta$ for Visual Decathlon (Rebuffi et al., 2017) and PACS (Li et al., 2017).

| $\eta$ | 0.05 | 0.1 | 0.2 | 0.3 | 0.4 | 0.5 |
|---|---|---|---|---|---|---|
| Visual Decathlon | 86.33 | 86.43 | 86.58 | 86.47 | 86.57 | 86.41 |
| PACS | 93.67 | 93.32 | 94.31 | 94.06 | 94.01 | 93.71 |

Table 7: Performance of WDT when inserted at different depths of ResNet26.

| Position | Before `block1` | Before `block2` | Before `block3` | Before FC |
|----------|-----------------|-----------------|-----------------|-----------|
| W-Acc.   | 86.58           | 84.28           | 80.07           | 35.45     |

Table 8: Sentiment classification accuracy for underrepresented latent domains.

| $d$ | Virgin | American | United | Delta | Southwest | US Airw. | W-Acc. |
|-----|--------|----------|--------|-------|-----------|----------|--------|
| $\pi$ | 0.034 | 0.188 | 0.261 | 0.152 | 0.165 | 0.200 | |
|       | 73.05 | 82.01 | 77.58 | 72.44 | 75.13 | 84.84 | 78.53 |
| Ratio | | 0.25 | | | 1.0 | | |
|       | 62.17 | 80.96 | 74.92 | 71.87 | 75.60 | 84.12 | 77.11 |

Moreover, WDT should be placed early in the network, as transferring information through augmentation at later stages of the model negatively affects performance, see Table 7.

SENTIMENT ANALYSIS

We construct an experiment on language data to showcase that latent domain suppression is not restricted to the realm of images. For this, we make use of a publicly available dataset that contains sentiment scores over Tweets addressed at publicly maintained airline accounts. The dataset is available at this url.

In this scenario, we identify $d$ with different carriers, e.g. Virgin, US Airways, etc. The experimental setup closely follows that in Section 4.4: first, we train a latent domain model on all available data and record its performance. Next, the first three hidden domains are subsampled (25% are kept), after which sentiment models are trained anew on the remaining data. We use a simple model here (ridge regression over tf–idf vectors) as the focus is to show that latent domains *are* suppressed in NLP problems — just as for images.

Accuracies are shown in Table 8. The largest drop in performance is registered on the smallest domain (Virgin) for a relative change in accuracy of -14.9%. At the same time, performance is stable on larger domains. As in previous experiments, a comparatively small drop in weighted accuracy (-1.8%) hides the severity with which small domains are suppressed by the model.

Table 9: Topic classification on AG News.

| Topic | Business | Science | Sports | World | W-Acc. | Unif.-Acc. |
|-------|----------|---------|--------|-------|--------|------------|
| $\pi$ | 0.1 | 0.1 | 0.4 | 0.4 | | |
| VDCNN | 85.7 | 85.2 | 90.4 | 96.8 | 92.0 | 89.5 |
| DRA | **87.9** | **85.6** | **91.5** | **97.1** | **92.8** | **90.5** |

TOPIC CLASSIFICATION

For our topic classification trials, we make use of VDCNN language models that resemble residual networks (Conneau et al., 2016). Due to their residual property, these can easily be extended with DRA. The main difference here is that convolutions are one-dimensional in VDCNN (applied to text data) versus ResNet (for images), a difference that is straightforward to account for in DRA.

Experiments are carried out over AG News which is widely used in scientific NLP publications.[5] To test model's robustness against underrepresented modes, we randomly drop 75% of examples from the "Business" and "Science" topic category. The results shown Table 9 compare the 9-layer VDCNN to the *exact same* model extended with our DRA ($K = 2$) at every layer. Models are optimized via SGD and 0.9-momentum for 100 epochs, with learning rate halvings each 10 epochs.

---

[5]See for instance `paperswithcode.com/sota/text-classification-on-ag-news` for an overview of models applied to AG News.

Table 10: Accuracies for ResNet18 and DRA on three medical image benchmarks.

| | ResNet18 | | DRA | | |
| | train | *test* | train | *test* | |
|---|---|---|---|---|---|
| DermaMNIST | 89.39 | 72.07 | 94.64 | **73.22** | +1.60% |
| OCTMNIST | 99.77 | 73.10 | 99.77 | **75.00** | +2.60% |
| RetinaMNIST | 71.76 | 49.50 | 84.82 | **53.00** | +7.07% |

MEDICAL IMAGE ANALYSIS

We evaluate DRA on images from the medical domain. As baseline we employ a ResNet18, and we follow the optimization settings (SGD, 0.9-momentum) outlined in Yang et al. (2020).

We evaluate models on (i) DermaMNIST, a 7-way classification problem that is constructed from HAM10000 (Tschandl et al., 2018); (ii) OCTMNIST contains optical coherence tomography images showing 4 retinal diseases based on Kermany et al. (2018); (iii) RetinaMNIST containing retina fundus images from DeepDRiD (https://isbi.deepdr.org/data.html).

We make no modifications to the benchmark data and use it as is, training both ResNet18 and the exact same model with DRA ($K = 2$) added. The results shown in Table 10 highlight that DRA provides a robust increase in test accuracy, a finding that is in line with previous results that indicated it enhances performance over multi-modal distributions (c.f. Table 3).

