# OpenReview forum: "Reducing Implicit Bias in Latent Domain Learning"
_ICLR.cc/2021/Conference — Reject_

### Official Review · AnonReviewer2 · 2020-10-25
**Impressive results on PACS (single task, multi-domain) for latent domain learning with a gate**

**Rating:** 6
**Confidence:** 2

**Review:**

1. Summarize what the paper claims to contribute. Be positive and generous.
The paper claims to contribute a new method *dynamic residual adapters (DRA)* coupled with *weighted domain transfer (WDT)* to tackle *latent domain learning.* The proposed method improves the model performance against small domain datasets without hurting the model performance against large domain datasets in two different settings:
  (1) multi-domain setting (10 different tasks with 10 different domain per task)
  (2) multi-style setting (1 task with 4 different styles)
Impressive empirical results! I especially enjoyed reading PCA graphs and its qualitative analyses.

2. List strong and weak points of the paper. Be as comprehensive as possible.
  (1) Strengths
    a. Exhaustive empirical analyses. (ablation tests and tests with varying K value)
    b. Qualitative analyses backed by graphs like PCA and example images.
    c. Impressive accuracy improvements.
    d. Intuitive theoretical explanation. Cool idea to use a gate to make RA dynamic.
  (2) Weaknesses
    a. Skipped the math that yields the equation (3) in the section 3. Would be nice if the steps are attached as an Appendix.
    b. In Table 1, there is a data domain size metric: $\pi_{d}$ . Can you add how this is computed? Also, how small is the smallest data in sheer number of examples?
    c. Table 1 has RestNet56, but Table 2 doesn't. Why did you make this choice of experiment design?
    d. With PACS dataset, you have experimented with the model (k-means+RA). I didn't quite understand the model setup and the motivation. In my understanding, the model learned the latent domain labels via k-means. And, then, based on this pseudo domain labels, the model is fine-tuned with Residual Adapter applied. Did I understand the model correctly? What is the motivation of doing this? I am not sure if this is a fair comparison between the RA and the proposed DRA+WDT methods. It seems rather a comparison between DRA+WDT and K-Means.
    e. In Figure 3, the paper says "Middle: WDT exchange or different domains, sketch is particularly inactive." I had a bit of difficulty parsing what "inactive" means here because the "Middle" figure is about "WDT Exchange Strength" and because "sketch" has the largest strength. Can you explain what "inactive" here means?
    f. Based on Table 3, the positive effect of WDT is not strong. I wonder if WDT is necessary.
    g. It seems to me that the strength of the proposed method is much more evident in the second problem type (PACS) where the task is the same across different domains. In the first problem type (Visual Decathlon), the DRA+WDT's performance boost is not consistent across different domains. I see that DRA+WDT hurts the performance compared to the baseline ResNet26/56 on a few different domains, such as Daim., Gtsrb, Omn., and Svhn. $\pi_{d}$ values of the domains of PACS are greater than those of Visual Decathlon, excluding svhn. Why do you think that is? When should one use or not use DRA+WDT in order to avoid hurting the model performance?
    h. Minor formatting issues. See 6 below.

3. Clearly state your recommendation (accept or reject) with one or two key reasons for this choice.
Accept because of the impressive performance of the proposed method against PACS (single task, multi domain) with clear visual analyses. Meantime, 2.(2).g requires more explanation to make the paper's claim stronger.

4. Provide supporting arguments for your recommendation.
See 2.(1) Strengths, 3, and 2.(2).g above.

5. Ask questions you would like answered by the authors to help you clarify your understanding of the paper and provide the additional evidence you need to be confident in your assessment.
See 2.(2) Weaknesses above.

6. Provide additional feedback with the aim to improve the paper. Make it clear that these points are here to help, and not necessarily part of your decision assessment.
  (1) The indentation of Table 5 seems to be inconsistent with the rest of the tables in the paper.
  (2) In Figure 3, the paper mentions $P_{k}$ without the denotation explained explicitly in any of the main body of the text. I had to re-read the paper to find a footnote 5 to finally understand what this denotation meant. It would be good to briefly explain this denotation in the same description of Figure 3.

---

> ### Author Response · Authors · 2020-11-20
> **Clarifications around WDT & ResNet56 experiments**
>
> Dear Reviewer, many thanks for your suggestions. We address them point-by-point below.
>
> *"Skipped the math that yields the equation (3) in the section 3."*
>
> As requested, we have reorganized this section to clarify those points. Please let us know in case any concerns remain!
>
>
> *"In Table 1, there is a data domain size metric: $\pi_d$. Can you add how this is computed? Also, how small is the smallest data in sheer number of examples?"*
>
> $\pi_d$ indicates the relative share of each (hidden) domain, i.e. we compute it as $N_d/N$, where $N_d$ denotes the number of examples belonging to latent domain $d$, and $N$ is the sum over all domains (we also mention this in the first paragraph of Section 3.1). Note that we only compute $\pi_d$ for the analysis – the model has absolutely no knowledge of which domains are large/small, nor how many there are.
>
> The smallest domains are: 1.6k (PACS-photo) and 2k (Vgg-Flowers).
>
>
> *"Can you explain what "inactive" here means?"*
>
> WDT is a pairwise augmentation, and the exchange strength is a measure of how similar each latent domain is. In other words: how much parameter sharing occurs between some latent domain (e.g. VD-Omniglot) and the other ones.
>
> For example: PACS-sketch is relatively isolated in feature space (see Fig. 3 left), and "inactive" here means the average exchange strength is relatively low, i.e. **a lower average δ in WDT = fewer parameters shared with other PACS domains = an inactive domain.**
>
> *"Table 1 has RestNet56, but Table 2 doesn't. Why did you make this choice of experiment design?"*
>
> That’s a great suggestion, we have added results for ResNet56 to Table 2. However, the improvement over ResNet26 is only marginal (92.70 -> 93.35) and significantly smaller than the improvement via DRA (92.70 -> 94.76). Note DRA also uses around 40% less parameters than ResNet56.
>
> This result once again confirms that **performance on small latent domains is suppressed**, and **adding more layers** – the preferred option for standard classification – **doesn’t solve this problem**.
>
> Please let us know of any concerns that remain, so that we may address them.

---

### Official Review · AnonReviewer3 · 2020-10-27
**interesting problem, but lack of new insights**

**Rating:** 4
**Confidence:** 4

**Review:**

Summary:
The authors propose a method for latent domain learning, where input data come from different domains and the domain labels are unknown. The proposed method consists of two parts: dynamic residual adapter and weighted domain transfer. The dynamic residual adapter acts as a mixture of expert layer. And the weighted domain transfer which augments the dataset by interpolating between different input pairs. Empirical results show that when combined together, the proposed method perform better than training a regular model.

Pros:
1. Latent domain discovery is a very interesting topic.
2. Empirical results show that the method brings improvement to minority domains.

Cons:
1. Maybe I missed something, but I don't there are new insights in the paper. The proposed dynamic residual adapter is just an instance of MoE [1] with adapters, which I think should be a baseline in the experiments.
2. "Section 3.4 Weighted Domain Transfer" is not well-motivated and very confusing. Here you want to interpolate between x_i and x_j. But why do you compute the difference between the input x_i and the feature \mu_i in equation 3? Are they comparable with each other? What is the goal that you want to achieve here?

Other comments:
1. I think the introduction describes the problem too much, leaving it no space to expand your idea and intuition. For example, you start describing your idea at the very last paragraph.
2. When creating the augmented examples, can you leverage the gate information that you produced from the MoE?
3. It will be more helpful to understand the DRA component if you can provide PCA over the original activations.


[1] OUTRAGEOUSLY LARGE NEURAL NETWORKS: THE SPARSELY-GATED MIXTURE-OF-EXPERTS LAYER

---

> ### Author Response · Authors · 2020-11-20
> **Clarification of novel contributions made & qualitative analysis**
>
> Dear Reviewer, thank you for your comments, which we address point by point below.
>
> *“Maybe I missed something, but I don't there are new insights in the paper. The proposed dynamic residual adapter is just an instance of MoE [1] with adapters, which I think should be a baseline in the experiments.”*
>
>
> We make several novel contributions in this paper.
>
> 1. The most important insight is the **identification of a new problem** that is pervasive yet not generally known by the community. Specifically, we present clear empirical evidence that **standard deep models underperform when latent domains are present due to suppressing small domains**. This is a *fundamental problem with the generalization of deep architectures*, and we have included results (page 9 of our revision) that demonstrate it even extends to standard classification problems, e.g. on CIFAR-10 (see Table 4).
>
> 2. We find that standard augmentation strategies do not remedy this problem. Because of this, **we propose WDT, a new augmentation strategy that reduces bias against small latent domains**.
>
> 3. As per MoE: this is simply an elegant way of extending multi-domain approaches to latent domains, where we cannot use off-the-shelf multi-domain methods (since there are no domain labels to speak of, see page 4 paragraph 3). Because there is no domain information, we **propose DRA, an alternative way to target corrections purpose-built for latent domain learning**. Please note DRA is not necessarily an instance of (I) MoE – we explored several alternatives, such as (II) Gumbel-softmax mechanisms that we compare to in Table 5, or (III) fixed assignments (e.g. after applying latent domain discovery e.g. through k-means, see Table 2).
>
>
> *"Here you want to interpolate between $x_i$ and $x_j$. But why do you compute the difference between the input $x_i$ and the feature $\mu_i$ in equation 3?"*
>
> We discuss the motivation around WDT on page 5 paragraph 2: for latent domains, we do not want to exchange between samples that are visually too similar. This has a straightforward motivation: exchanging too much information between samples *can* reduce the number of discriminative features for samples from the same class. What we want instead is to encourage information exchange between different (hidden) domains, so as to end up with a model that learns concepts of objects (cats, dogs, etc.) that are invariant to the (latent) domain.
>
> Please note the difference $x_i-\mu_i$ is further scaled by the standard deviation after which $\mu_j$ is added. This process is crucial and exactly responsible for the *transfer of statistical information from $x_j\to x_i$*.
>
>
> *"I think the introduction describes the problem too much, leaving it no space to expand your idea and intuition. For example, you start describing your idea at the very last paragraph."*
>
> We maintain a longer than usual introduction is needed, because an important aspect in this paper is to **establish latent domain learning as a novel learning setting**. This requires we explain the latent domain scenario in sufficient detail, in large part so that future work won’t have to. Feel free to check out the long introduction in the initial work on multi-task learning by e.g. Caruana (1997). While it might be a bit obvious nowadays, it is still recommended literature for anyone that is starting out in the field.
>
>
> *"When creating the augmented examples, can you leverage the gate information that you produced from the MoE?"*
>
> We agree this is an interesting idea, as having access to richer conditional information within DRA could potentially enhance the internal clustering of latent domains. However, in doing so the method would no longer be end-to-end, as we would require a full pass through the network to collect gate activations. The goal of our manuscript – as stated on page 2 paragraph 1 – is to contribute a principled end-to-end mechanism that doesn’t have a large negative impact on runtimes (e.g. we want to avoid 9x models). Requiring two passes would have moved the paper in a very different direction.
>
>
> *"It will be more helpful to understand the DRA component if you can provide PCA over the original activations."*
>
> Please note in Figure 2 we do not display PCA over activations. We collect *only the gate activations* across the ResNet26, and then reduce this to two dimensions with a linear mechanism (i.e. PCA). This shows:
> * individual residual adaptations *can* be attributed to latent domains.
> * **DRA shares between similar domains**, which provides an **explanation** of its much better performance than RA etc.
>
> We went ahead and added some clarification around this in the final paragraphs of page 6.
>
> Please let us know in case there’s anything else you would like us to revisit.

---

> > ### Comment · AnonReviewer3 · 2020-11-25
> > **reply to the rebuttal**
> >
> > Thanks for your responses.  I do strongly agree that the setting is very interesting and practical. However, I still have several comments:
> >
> > 1. I found the motivation around WDT still unclear. You mentioned that "for latent domains, we do not want to exchange between samples that are visually too similar". But then what will the interpolation be like if you are exchanging information from two very distinct images? Also in cases where you have distinct label set for different latent domains, learning concepts of objects (cats, dogs) that are invariant to the domain is not well-defined.
> >
> > 2. For the PCA part, what I was thinking is that in the case where the label set of different latent domains are distinct, their input distributions are usually very disjoint as well (for example, images from Omniglot should look very different from aircraft). In this situation, the original ResNet may already give you a separated embedding space. Applying clustering over this embedding space and run distributionally robust optimization may give you a very strong baseline.
> >
> > 3. I still think DRA is an instantiation of MoE.

---

> > > ### Author Response · Authors · 2020-11-25
> > > **Second response, PCA for ResNet26**
> > >
> > > > I found the motivation around WDT still unclear. You mentioned that "for latent domains, we do not want to exchange between samples that are visually too similar". But then what will the interpolation be like if you are exchanging information from two very distinct images? Also in cases where you have distinct label set for different latent domains, learning concepts of objects (cats, dogs) that are invariant to the domain is not well-defined.
> > >
> > > One of the challenges in latent domain learning is *domain leakage*: in multi-domain learning there are domain annotations, so the probability of classifying some f(x) to a wrong domain is *zero*. Remember, this is *not* the case here.
> > >
> > > This is exactly where WDT helps: by interpolating between domains the model learns to look at more than just domain-specific (low-level) features, and becomes focused on the (high-level) semantic features relevant for object classification.
> > > What we mean with "we do not want to exchange between samples that are visually too similar": we simply point out that if you exchange too aggressively, you eventually end up removing features that *are* crucial to detecting semantic categories.
> > >
> > > > For the PCA part, what I was thinking is that in the case where the label set of different latent domains are distinct, their input distributions are usually very disjoint as well (for example, images from Omniglot should look very different from aircraft). In this situation, the original ResNet may already give you a separated embedding space.
> > >
> > > Thanks for the clarification, we understand your request better now. We have added a PCA over ResNet26 for VD in the Appendix. As one can see there, indeed Omniglot is singled out (due to its low-level differences). Higher level concepts are however not picked up on by the ResNet, which *otherwise exhibits no latent domain structure*.
> > >
> > > Again, note that our main goal is not to identify the original domains but to automatically find the optimal latent domains for our learning objective, i.e. classification. Think of a case where two domains are *annotated as different*, but are actually from the *same distributions* – traditional multi-domain models will not be efficient in this setting!
> > >
> > > Furthermore, we already show that using the ground-truth domain information (RA in Table 2) or using a clustering method (K-means in Table 2) obtains lower performance than our end-to-end methodology.
> > >
> > > Fig. 2&3 show that the gating output encodes discriminative information about the original domains. In particular, they provide evidence that models *separate latent domains internally* when learning end-to-end – i.e. models "agree" that domain separation makes sense (they *could* simply ignore latent domains, after all), however we can also see that parameter sharing between *some* domains is crucial to maximizing model performance.
> > >
> > > > I still think DRA is an instantiation of MoE.
> > >
> > > We never disagreed, as MoE is an *extremely general concept*. Note however that any self-conditional attention mechanism – from a formal viewpoint – can be viewed as a modular instantiation of MoE. We believe that research (where appropriate) should make connections between current ideas & past ones. Pointing out a *clear* connection between MoE and self-attention is doing just that.
> > >
> > > We thank you for your second response and your suggestions, which have been very helpful in improving this manuscript.

---

### Official Review · AnonReviewer1 · 2020-10-28
**Review #1**

**Rating:** 5
**Confidence:** 4

**Review:**

------ Update after discussion with authors ---------

I would like to thanks the author for their efforts by adding additional experiments, which surely enhances the significance of the proposed approach. Based on these, I increased my score to 5.

I have re-checked the final revised version, I think the current version still *requires proper organizations and justifications*. For example, the added experiments still talked about the accuracy, the in-depth analysis seems lacking. I think a substantial revision of the paper in terms of structure, idea presentation, and analysis is still needed. Based on this, my final score is 5.

-----------------------------------
Summary:

This paper studied how to learn a neural network with multiple domains without knowing the exact domain label (by merging all the domains as a large domain). Then they proposed dynamic residual adapters and weighted domain transfer to address this issue. The empirical results showed its practical benefits.

------------------------------------------------------

Overall review

Pros:

[1] This paper is well-motivated. I like the analyzed scenario and I think it can have a strong practical utility.

[2] The high level of proposed ideas is technically sound.

Cons:

[1] The submitted version seems to be a preliminary version with many missing and unclear elements.

[2] As an **empirical** paper, the experimental results are not sufficient for ICLR.

[3] Some technical details need better justifications and discussions.

Based on these, I recommend a rejection at this time but encourage a major revision for resubmission.

----------------------------------------------------

Detailed explanations

[1] Missing elements

[a] I am rather confused and unclear about the whole learning procedure. It seems the author used DRA in the residual module. However, the role of WDT is unclear. What is the global training loss in the proposed approach? WDT is a part of the loss or used for analyzing the problem?  I would like to see a pseudocode/protocol for the whole algorithm or a clear network structure for illustrating the idea in Sec 3.3-3.4.

[b] The mathematical notations defined in this paper are presented oddly (particularly in sec 3) for example:

[b1] Equation (1), $\alpha$ and $\theta$ are scalars or vectors? what is meaning for $|\alpha|\ll |\theta|$? I guess it is $\text{dim}(\alpha)$ but it makes me rather confused.

[b2] The same problem for eq(2) and $\epsilon$

[b3] In WDT, the same problem for $\delta$, $\delta_i$ and $\delta_j$

These confusions make it more difficult to understand the approach.

[2] The empirical results

The current empirical results only compare MLFN, which is not sufficient.

I noticed the author claimed, “Note the goal here is not to compare to the performance of existing multi-domain approaches that Visual Decathlon was designed for, but to show that deep networks struggle with learning small latent domains when no domain annotations are provided.”

I agree with this opinion if the paper aims to only analyze this scenario (generally from a theoretical perspective).  These kinds of experiments are sufficient.

By contrast, the current version aims at **proposing a new empirical approach for the real-world practice**, which is not sufficient. I would like to see a **strong practical result** either outperforming the recent baselines or applying in many real-world problems.

[3] Technical details

[a] I suggest not using the term “domain labels” since it can be confusing to label $y$ information in the unsupervised domain adaptation. I think “domain index” or “task index” are better choices.

[b] The visualization of $\delta$ sounds interesting but I can not understand the meaning. A better explanation is expected.

[c] Fig (1),(2) why PCA visualization? Why not Tsne?

[d] The benefits of self-attention are unclear. More analysis (not numerical accuracy) is expected.

--------------------------------------------

Suggestions

I suggest a major revision on the proposed approach, empirical results, and more analysis (not accuracy) on the benefits of the idea.

---

> ### Author Response · Authors · 2020-11-20
> **Clarifications around WDT, latent domain learning & revision of notation**
>
> Dear reviewer, many thanks for your review. We go through your comments point by point.
>
> *“I am rather confused and unclear about the whole learning procedure. [.. ] WDT is a part of the loss or used for analyzing the problem?”*
>
> We have pointed this out at various points throughout the paper – *including* the abstract: WDT is an augmentation strategy. Just like other strategies (grayscaling, flipping, or MixUp) this does not explicitly appear in the loss function. WDT is needed in latent domain learning because strategies that work well for standard classification problems (such as MixUp) do not work here (see Table 5).
>
>
> *“The mathematical notations defined in this paper are presented oddly [..]”*
>
> We follow standard ML notation for all parts of our paper, distances δ for example are always scalar by their definition; ϵ is a channel-sized vector (as we state on page 4). At your request, we did go ahead and clarified notation further in our revision, e.g. replaced |**x**| with dim(**x**), and have introduced bold symbols for vector-valued objects such as **x**.
>
>
> *“The current empirical results only compare MLFN, which is not sufficient [..] I would like to see a strong practical result [..] outperforming recent baselines or applying in many real-world problems.”*
>
> As we state on page 7 paragraph 4, there currently exist no customized end-to-end solutions for latent domain learning in the literature. If there is any approach you feel we’ve overlooked here, please point them out.
>
> Moreover, the goal of this paper is very clear: to **make a *first* step toward models that learn reliably in the presence of latent domains**, by preserving small latent domains in the dataset. Contrary to other work that expands on previous learning settings (multi-domain, domain adaptation, etc.), this requires a significant extra effort in terms of motivating the problem. We do this here through an extended introduction, and furthermore present multiple qualitative insights. It is **standard protocol** in the multi-domain literature (compare Rebuffi et al. (2017, 2018), Guo et al. (2019), Liu et al. (2019), etc.) to use datasets like PACS or Visual Decathlon as a proxy for the real-world. As such, the criticism that there is no “real world experiments” has to be put into perspective – we are not using toy data after all, and e.g. Visual Decathlon is a **very** complex dataset.
>
>
> *“The visualization of δ sounds interesting but I can not understand the meaning. A better explanation is expected.”*
>
> WDT is a pairwise augmentation. This figure visualizes how much exchange occurs on average by each latent domain. In other words: how much parameter sharing happens between some latent domain (e.g. VD-Omniglot) and the other ones.
>
> For example: some domains (e.g. PACS-sketch) are quite isolated in feature space (see Fig. 3 left), and we show that this directly translates to a **lower average δ in WDT = less parameter sharing with other PACS domains**. These visualizations confirm that WDT does what it was designed for.
>
>
> *“Fig (1),(2) why PCA visualization? Why not Tsne?”*
>
> Because when features are separable by the most simple, *linear* unsupervised mechanism (which is PCA, not the *nonlinear* t-SNE), then this is a much stronger statement that there is *meaningful separation* within DRA.
>
>
> *“The benefits of self-attention are unclear. More analysis (not numerical accuracy) is expected.”*
>
> We firmly establish the benefits of self-attention. We analyze alternatives to self-attention in Table 5, and in particular find that Gumbel-softmax negatively impacts performance. We also compare to K=1, i.e. having no self-attention.
>
> We hope we were able to remedy your concerns in our updated revision. If there’s anything else you would like us to revisit, please let us know.

---

> > ### Comment · AnonReviewer1 · 2020-11-21
> > **Response for the rebuttal**
> >
> > Thanks for your responses. It surely clarifies my several confusions!  However, I still have several questions that require properly addressed in the paper. The following are my comments:
> > > WDT is an augmentation strategy
> >
> > Thanks for your clarification! The following questions still exist
> >
> > 1. *What is the global training loss in the proposed approach*
> >
> > 2. *a pseudocode/protocol for the whole algorithm or a clear network structure for illustrating the idea in Sec 3.3-3.4.*
> >
> > I think this is quite important for understanding your approach. e.g. the step-by-step description of the proposed approach. Since this paper currently does not provide source code, this is particularly important.
> >
> > > clarified notation further in our revision
> >
> > Thanks, I have checked the revised version. It is much better.
> >
> > >  to make a first step toward models that learn reliably in the presence of latent domains
> >
> > Thanks for your remark. But as an *empirical paper*, the current empirical studies are indeed not sufficient for ICLR. For example, is your approach still valid in NLP dataset? Current results mainly focus on image problems *on the benchmark*.  I would like to see a real-world application such as medical image.
> >
> > Alternatively, the authors can provide a theoretical analysis of the proposed approach, to enhance the contribution of the paper.

---

> > > ### Author Response · Authors · 2020-11-24
> > > **Second response, new NLP & medical imaging experiments**
> > >
> > > Thank you for your quick response, which we much appreciated!
> > >
> > > > 1. What is the global training loss in the proposed approach
> > > > 2. a pseudocode/protocol for the whole algorithm or a clear network structure for illustrating the idea in Sec 3.3-3.4.
> > >
> > > Because there are no domain labels, global training occurs via standard cross-entropy. This is the very point of designing an end-to-end method, as we can therefore optimize our methods using standard minimization of $\nabla\big(\sum_i L(y_i,f(x_i))\big)$.
> > >
> > > The only important bit to watch out for is that (I) images are modified via WDT (because we are dealing with latent domain problems here) and (II) because we have DRA, we only optimize the model weights $\alpha$.
> > >
> > > > a pseudocode/protocol for the whole algorithm or a clear network structure for illustrating the idea in Sec 3.3-3.4.
> > >
> > > We have added pseudo-code at your request (see Algorithm 1, page 14) and will release source code and models alongside a final version. As this makes clear, our methods are of general purpose and **can be combined easily with existing methods** – this aspect was a central motivation to our work.
> > >
> > > > Current results mainly focus on image problems on the benchmark.
> > >
> > > The use of these benchmarks is really complete standard protocol. Exactly the same benchmarks are used in recent works in the multi-domain literature:
> > > * Rebuffi et al. (2017, 2018) NeurIPS & CVPR.
> > > * Guo et al. (2019) AAAI.
> > > * Liu et al. (2019) CVPR.
> > >
> > > > For example, is your approach still valid in NLP dataset?
> > >
> > > We have added two new results to address this question:
> > > * Sentiment analysis on airlines Tweets (Table 8) under shrinking domain sizes. Main result: we can again correlate smaller domains directly with a drop in performance, just as for images.
> > > * Topic classification with VDCNN (baseline) and DRA inserted into the network (Table 9). We follow the experimental setup in Section 4.4 and subsample 2 out of 4 topics to create an unbalanced version. **DRA improves over VDCNN**
> > >
> > > The above results confirm that there *exists no reason to believe that latent domain suppression only occurs in images*. That being said, as the absence of any modular/architectural solutions for latent domains in the NLP literature indicates, adding small domain robustness to NLP models is a research question in it’s own right that requires significant effort (for an indication of this see e.g. the NLP multi-task work of Stickland & Murray (2019)), but this is something we hope to address adequately in future work!
> > >
> > > > I would like to see a real-world application such as medical image
> > >
> > > We followed your suggestion and have added results for coupling our approach with a ResNet18 (Table 10) on medical images, where we show that DRA yields a *robust increase in performance on multi-modal medical image data*.

---

### Official Review · AnonReviewer4 · 2020-11-04
**This paper is well written. The motivation of this paper is clear and the proposed framework does break the limitation of the existing deep learning methods.**

**Rating:** 6
**Confidence:** 3

**Review:**

The paper describes dynamic residual adapters designed to adaptively account for latent domains, and weighted domain transfer. This framework injects adaptivity into networks, preventing them from overfitting to the largest domains in distributions, a failure mode of traditional models that are exposed in latent domain learning. The approach closes a large amount of the performance gap to domain-supervised solutions.

This paper is well written. The motivation of this paper is clear and the proposed framework does break the limitation of the existing deep learning methods. Below I present some suggestions, which hopefully can help the authors improve their study:

The authors do not describe any processing they have done of the data. This should be clearly included in the methods section. More experimental details and insightful discussions should be provided. I suggest the authors repeat the benchmarking using a selection of datasets more similar to those used in current studies.
The benchmarking results are insufficiently described in the text and can only really be seen in the tables/figures. The authors should explain the results in more detail in the text and include their interpretation of the results.
Section 3. In this part, the authors introduce the algorithm. Since there are many formulas in this section, additional explanations on the learning procedure would help understand the proposed method.

---

> ### Author Response · Authors · 2020-11-20
> **Included additional experiments & clarifications**
>
> Thank you for your constructive feedback. We address each point below.
>
> *“The authors do not describe any processing they have done of the data. This should be clearly included in the methods section.”*
>
> We do discuss this on page 6 paragraph 3. As we mention there, we only use standard augmentations (flipping, normalization) in *all* of our experiments, and otherwise make no changes to the benchmark data.
>
>
> *"The authors should explain the results in more detail in the text and include their interpretation of the results."*
>
> We extensively discuss our results, and provide a large set of experiments for two latent domain settings, with (I) joint and (II) disjoint label spaces. Moreover, for both settings, we provide (III) qualitative insights, as well as (IV) thorough discussions of the computational requirements. Please note the 8 page limit prevented us from including additional analysis in the main paper, but we included an extensive study of DRA around (V) single dataset performance on CIFAR-10 and CIFAR-100 (Table 3), (VI) robustness to class imbalance (Table 4), as well as (VII) ablations (Tables 5-8) in the appended pages.
>
> We have taken advantage of the 9 page limit for the revision, and following your suggestion included additional analysis (V)+(VI) (which were previously in the appendix) in the main parts of the paper. On page 9 we demonstrate that the benefits of DRA even extend to standard classification tasks, and DRA brings *significant* performance advantages when some classes are underrepresented in the data. The last case addresses a particularly severe issue for standard deep learning models, as we display in Table 4 for ResNet26+56.
>
> *"In this part, the authors introduce the algorithm. Since there are many formulas in this section, additional explanations on the learning procedure would help understand the proposed method."*
>
> We have reorganized the section, and added additional discussions around WDT in our revision.
>
> Thank you again for your constructive feedback. Please point out any remaining concerns so that we may address them.

---

### Author Response · Authors · 2020-11-24
**Overview of changes for rebuttal**

We have made use of the extended 9 page limit for the rebuttal and incorporated content that was previously discussed in the Appendix:
* We highlight that latent domain suppression is a *core problem for ML generalization*. Our experiments with undersampled CIFAR-10 classes (Table 4) show that standard models (ResNet26, ResNet56) struggle to generalize in standard classification settings. We show DRA is a very effective counter against this.
* Added results for single datasets. DRA improves performance, especially for highly varied (multi-modal) datasets such as CIFAR-100. This makes sense as a highly modal dataset is – under its surface – a latent domain problem.
* Revised notation and motivation for WDT, which should now be much clearer.
* Added results for PACS with ResNet56. It’s poor performance shows: *deeper models are not a remedy for latent domain suppression*, which instead requires customized solutions like DRA and WDT.

In addition, we added new results for:
* **NLP**- demonstrating that latent domain suppression is a *general problem for different types of data*. We couple DRA+VDCNN and show it boosts performance in topic classification.
* **Medical images**- showing DRA boosts performance on multi-modal medical image benchmarks.

---

### Decision · Program_Chairs · 2021-01-07
**Final Decision**

**Decision:**

Reject

**Comment:**

While all reviewers agree that the topic is interesting and the work has merit, several issues have been pointed out, especially by R1 and R3, that indicate that the work is not  ready for acceptance at this stage. the authors are strongly encouraged to continue to work on this topic, taking into account the feedback received.